# Quantification of early nonpharmaceutical interventions aimed at slowing transmission of Coronavirus Disease 2019 in the Navajo Nation and surrounding states (Arizona, Colorado, New Mexico, and Utah)

Ely F. Miller[1], Jacob Neumann[1¤], Ye Chen[2], Abhishek Mallela[3,4], Yen Ting Lin[5], William S. Hlavacek[4], Richard G. Posner[1]*

1 Department of Biological Sciences, Northern Arizona University, Flagstaff, Arizona, United States of America, 2 Department of Mathematics and Statistics, Northern Arizona University, Flagstaff, Arizona, United States of America, 3 Department of Mathematics, University of California, Davis, California, United States of America, 4 Theoretical Division and Center for Nonlinear Studies, Los Alamos National Laboratory, Los Alamos, New Mexico, United States of America, 5 Computer, Computational and Statistical Sciences Division and Center for Nonlinear Studies, Los Alamos National Laboratory, Los Alamos, New Mexico, United States of America

¤ Current address: Department of Chemistry and Chemical Biology, Cornell University, Ithaca, New York, United States of America
* richard.posner@nau.edu

## Abstract

During an early period of the Coronavirus Disease 2019 (COVID-19) pandemic, the Navajo Nation, much like New York City, experienced a relatively high rate of disease transmission. Yet, between January and October 2020, it experienced only a single period of growth in new COVID-19 cases, which ended when cases peaked in May 2020. The daily number of new cases slowly decayed in the summer of 2020 until late September 2020. In contrast, the surrounding states of Arizona, Colorado, New Mexico, and Utah all experienced at least two periods of growth in the same time frame, with second surges beginning in late May to early June. Here, we investigated these differences in disease transmission dynamics with the objective of quantifying the contributions of non-pharmaceutical interventions (NPIs) (e.g., behaviors that limit disease transmission). We considered a compartmental model accounting for distinct periods of NPIs to analyze the epidemic in each of the five regions. We used Bayesian inference to estimate region-specific model parameters from regional surveillance data (daily reports of new COVID-19 cases) and to quantify uncertainty in parameter estimates and model predictions. Our results suggest that NPIs in the Navajo Nation were sustained over the period of interest, whereas in the surrounding states, NPIs were relaxed, which allowed for subsequent surges in cases. Our region-specific model parameterizations allow us to quantify the impacts of NPIs on disease incidence in the regions of interest.

**Data Availability Statement:** All data available on GITHUB: https://github.com/lanl/PyBNF/tree/master/examples/Miller2022NavajoNation.

**Funding:** YC, WSH, EFM, JN, and RGP acknowledge support from the National Institute of General Medical Sciences of the National Institutes of Health (grant R01GM111510). AM acknowledges support from the 2020 National Science Foundation Mathematical Sciences Graduate Internship Program and the Center for Nonlinear Studies at Los Alamos National Laboratory. WSH and YTL acknowledge support from the Laboratory Directed Research and Development Program at Los Alamos National Laboratory (project 20220268ER). We acknowledge use of the Monsoon computer cluster at Northern Arizona University, which is funded by Arizona's Technology and Research Initiative Fund to YC. The funders had no role in study design, data collection and analysis, decision to publish, or preparation of the manuscript.

**Competing interests:** The authors have declared that no competing interests exist.

## Introduction

An outbreak of pneumonia of unknown cause starting in Wuhan, China was recognized in late December 2019 and widely reported in early January 2020 [1,2]. The disease was later named Coronavirus Disease 2019 (COVID-19) [1]. The causative agent was identified as a novel coronavirus, later named Severe Acute Respiratory Syndrome Coronavirus 2 (SARS-CoV-2). COVID-19 rapidly spread to other countries [1–3], and the World Health Organization (WHO) declared the COVID-19 outbreak a pandemic on March 11, 2020 [4]. In the United States (US), during the early months of the pandemic, two regions were severely affected as measured by cumulative number of COVID-19 cases per capita: *Diné Bikéyah*, more commonly known as the Navajo Nation, and New York City.

On May 18, 2020, the Navajo Nation had the highest cumulative number of COVID-19 cases per hundred thousand in the US (2,344 cases per 100,000 residents), surpassing the New York City metropolitan statistical area (MSA), which had 1,806 cases per 100,000 residents [5,6]. Remarkably, both regions significantly slowed the transmission of COVID-19 and prevented additional surges in new COVID-19 cases until late September 2020 while many other regions, including the states of Arizona, Colorado, New Mexico, and Utah, each experienced a series of two surges in cases during the same period [6].

The President of the Navajo Nation and the Governors of the surrounding states of Arizona, Colorado, New Mexico, and Utah independently issued guidance and mandates to control the spread of COVID-19. These non-pharmaceutical interventions (NPIs) enforced or encouraged an array of behaviors that putatively protect susceptible individuals from SARS-CoV-2 infection, such as curtailing of travel, reduction in face-to-face interaction, face mask-wearing, working from home, etc. Although the governmental actions across the five regions were similar, there were notable differences, particularly in the duration of mandates, as we will discuss later. The disease transmission dynamics in the five regions were also different. In the Navajo Nation, the number of new cases detected daily rose sharply in late March to early April, peaked in May, and then steadily declined until late September 2020 [5]. In contrast, in each of the surrounding states, there were at least two periods of growth in new cases between 01-March-2020 and 14-September-2020 [6].

The objective of this study is to quantify the effect of early non-pharmaceutical interventions (NPIs) on the transmission of COVID-19 in the Navajo Nation (NN) and surrounding states: Arizona (AZ), Colorado (CO), New Mexico (NM), and Utah (UT). Insights into why the Navajo Nation experienced only a single period of growth in new COVID-19 cases during the period of interest while neighboring regions experienced two distinct phases of increasing case counts could point to strategies for controlling transmission of diseases similar to COVID-19 in the future. A possible explanation for the difference between the Navajo Nation (with one phase of growth in disease incidence) and the surrounding states (each with at least two phases of growth in disease incidence) is that NPIs were more effective and/or more sustained in the Navajo Nation. To evaluate this hypothesis, for each of the five regions of interest, we sought to use region-specific daily case reporting data to infer parameters of a compartmental model for COVID-19 transmission that accounts for subpopulations of susceptible individuals protected or not from SARS-CoV-2 infection by NPIs. The model structure allows for multiple phases, or periods, of NPIs. Each phase is associated with three parameters: an onset time, a sum of rate constants that defines a timescale for transition to a setpoint level of adoption of disease-avoiding behaviors, and the setpoint (the fraction of the regional population adopting disease-avoiding behaviors). We have previously shown that this model is able to reproduce the dynamics of regional COVID-19 epidemics in the 15 most populous metropolitan areas [7], including 2020 in 280 of 384 MSAs in the US [8], and in all 50 states [9].

Various studies [10–14] have evaluated the effects of NPIs in different regions of the world. The general conclusion is that NPIs limit COVID-19 transmission. Here we compared NPIs that were independently mandated in five regions of interest. Among these regions, the NN had a temporal profile of disease incidence that was qualitatively distinct. Here we investigate the possibility that this difference arose from NPI-related differences.

## Methods

### Study design, populations and setting

In a cross-sectional study, we used regional daily COVID-19 case data collected from January 21, 2020, to September 14, 2020, a compartmental model (Fig 1), and Bayesian inference to quantify time-varying NPI effectiveness for five different populations, those of the Navajo Nation and the four surrounding states (Arizona, Colorado, New Mexico, and Utah). In our model, NPI effectiveness relates to the fraction of a population partially protected against infection and less prone to transmit disease because of adherence to NPIs. This fraction is given by a step function $P_\tau(t)$ in the model.

The entire communities of the Navajo Nation and the four surrounding states were considered in this study. We considered disease prevalence in these five regions. Disease prevalence could be affected by age, household size, income/socioeconomic factors, population density, and occupational exposure. These factors were not considered in the study.

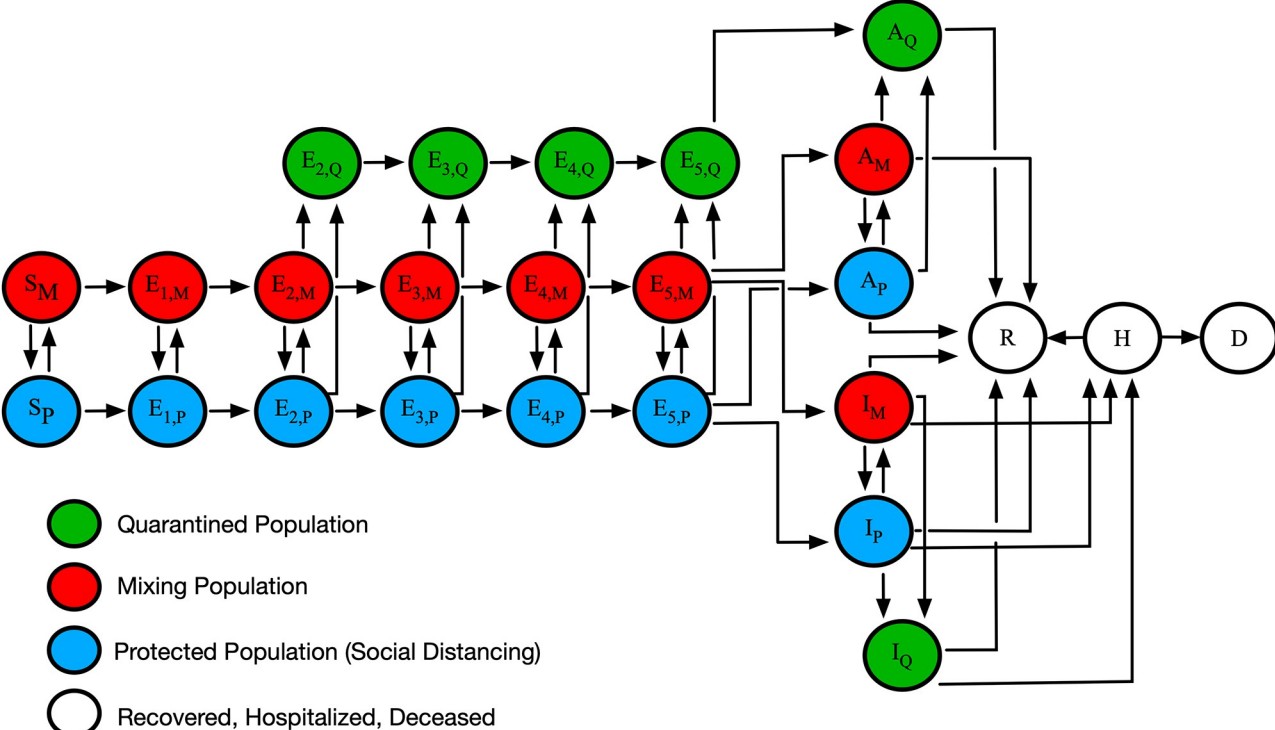

**Fig 1. An illustration of the mechanistic compartmental model used to analyze COVID-19 data** *(7)*. The model captures various subpopulations, as indicated in the legend. Transitions between subpopulations marked by M, P and Q subscripts represent adoption and relaxation of disease-avoiding behaviors. The model accounts for susceptible persons (S), exposed persons not experiencing symptoms while incubating virus (E), asymptomatic persons in the immune clearance phase of infection who never develop symptoms (A), infected persons with mild symptoms (I), infected persons with severe illness (H), deceased persons (D), and recovered persons (R). The incubation period is divided into five stages. Red (subscript M) indicates persons in the mixing population, blue (subscript P) indicates persons in the protected population, green (subscript Q) indicates persons in the quarantined or self-isolated population, and white indicates persons who are recovered, hospitalized, or deceased.

Comparison of NPI effectiveness in the Navajo Nation to that of the surrounding states was motivated by several considerations. The Navajo Nation experienced the highest per capita COVID-19 incidence in the US early during the pandemic [5]. Moreover, the time profile of disease incidence was qualitatively distinct from that of each of the four surrounding states, and the NPI mandates for the Navajo Nation were different from those of the surrounding states. Mandates were different partly because they were issued by the president of the Navajo Nation independently of state governors. Even if the mandates had been identical, the effectiveness of NPIs can vary depending on social structures. The Navajo Nation has unique social structures compared to the surrounding states. The Navajo Nation has one of the lowest population densities within the US, approximately seven persons per square mile. Persons of the Navajo Nation primarily live in clusters, where many individuals share the same household. Thus, despite low population density over the entire geographical area, household density can be quite high [15].

## Data collection and sources

We used public data extracted from newspapers and online resources. The COVID-19 surveillance data used to parameterize the model for the Navajo Nation were obtained from the *Navajo Times* COVID-19 webpage [5]. The *Navajo Times* provided daily reports of new confirmed COVID-19 cases over the period of interest. The reported source of this information was the Navajo Nation Department of Health (NNDOH) [16]. The COVID-19 surveillance data used to parameterize the models for Arizona, Colorado, New Mexico, and Utah were obtained from a GitHub repository maintained by *The New York Times* newspaper [17]. This GitHub repository collects new case reports from local health agencies in the United States. For each of the four states, we aggregated county-level case counts to obtain state-level case counts. In all regions of interest, positive cases were assessed according to CDC guidelines [18].

Regional NPIs were set by the president of the NN and the governors of AZ, CO, NM, and UT. Information about NPI mandates in the four states were obtained from the *John Hopkins Coronavirus Resource Center* [19]. This NPI resource webpage collects state-wide NPI mandates issued by each US state's governor and plots when they were issued against daily new cases to visualize the effect of NPIs on trends in new COVID-19 cases. The *John Hopkins Coronavirus Resource Center* collects policy data from various state-specific websites such as state and governor websites and from the National Governors Association. It should be noted that information in this resource characterizes state-level mandates only; information about county-level mandates is less readily available and was not considered in this study. Information about NPI mandates in the Navajo Nation was obtained from the NNDOH public website [16]. For our study we considered 4 types of NPIs: weekend lockdown, closure of nonessential businesses, mask mandates, and limitations on mass gatherings.

## Study bias

Information bias is a concern because the number of regional cases depends on the region-specific surveillance effort. To help account for information bias, we assumed that the detected fraction of cases is region-specific and constant over the time frame of interest. For each region, we estimated a parameter in our model $f_D$, the detected fraction of cases.

## Model and simulations

The model we considered in this study is that of Lin et al. [7]. It is a compartmental model that divides a regional population of interest into susceptible (S), exposed (E), infectious (I), and

removed (R) compartments (Fig 1). Exposed persons transition through a series of five stages, introduced to capture the distribution of incubation times observed for COVID-19 [20]. The model also accounts for quarantine, self-isolation because of symptom awareness, hospitalization, and death. Importantly, persons are allowed to transition between two modes of behavior, in which they are either protected (imperfectly) from infection (because of adoption of disease-avoiding behaviors) or are mixing freely (i.e., taking no special precautions to prevent infection). The model tracks 25 compartments. Each compartment corresponds to an ordinary differential equation (ODE). There is an auxiliary 1-parameter measurement model, which relates the variables of the compartmental model to reported new cases through surveillance testing [7]. The equations of the mechanistic compartmental model and the measurement model can be found in Appendix 1 of Lin et al. [7].

The model accounts for an initial phase of NPIs beginning at time $t = \sigma$, where $\sigma$ is fixed to the date the region of interest accumulated at least 200 COVID-19 cases. The model can be extended to account for $n$ additional periods. Thus, the total number of NPI periods considered in a regional model is given by $n + 1$ [7]. The start of a new NPI phase is accompanied by step changes in the values of the three NPI parameters. In the models for the Navajo Nation (NN), Arizona (AZ), Colorado (CO), New Mexico (NM), and Utah (UT), we considered three possible settings for $n$ (the number of additional NPI periods beyond the initial period): $n = 0$ (only one NPI period over the entire period of interest), $n = 1$ (two NPI periods), and $n = 2$ (three NPI periods). The setting for $n$ was determined as described below.

To determine the structure of the compartmental model for each region of interest (i.e., the number of distinct NPI phases), we used a heuristic model-selection method. In this approach, we calculated the value of the Akaike information criterion corrected for small sample size ($AICc$) for $n$ and $n + 1$ versions of the model, where $n = 0, 1$. We also calculated the value of the Bayesian information criterion ($BIC$) for the same two versions of each model. $\Delta AICc$ is defined as the change in $AICc$ between $n$ and $n + 1$ versions of the models: $\Delta AICc = AICc^n - AICc^{n+1}$. $\Delta BIC$ is defined similarly: $\Delta BIC = BIC^n - BIC^{n+1}$. We adopted $n + 1$ over $n$ when both of the following conditions held true: $\Delta AICc > 10$ and $\Delta BIC > 10$. The method of model selection described above was used to decide between the use of $n = 0$ and $n = 1$, and between the use of $n = 1$ and $n = 2$ [7,21].

In the case of only an initial NPI period ($n = 0$), the compartmental model and auxiliary measurement model have 20 parameters combined. Five of the parameters ($t_0$, $p_0$, $\lambda_0$, $\beta$, and $f_D$) are considered adjustable; these parameters are all region dependent. The other 15 parameters are taken to have fixed values. The 15 fixed parameters are $S_0$, $\sigma$, $I_0$, $m_b$, $\rho_E$, $\rho_A$, $k_L$, $k_Q$, $j_Q$, $f_A$, $f_H$, $f_R$, $c_A$, $c_I$, and $c_H$. The parameter $S_0$ represents the total population of the region of interest, as determined by census data [22], which we took to be fixed. $I_0$ refers to the starting number of infected individuals. We used $I_0 = 1$. $\rho_E$ and $\rho_A$ refer to the relative infectiousness of exposed persons and asymptomatic persons, respectively, compared to symptomatic persons [23,24]. Infected persons are taken to enter quarantine with rate constant $k_Q$ and persons with symptoms and mild disease are taken to self-isolate with rate constant $j_Q$. Persons in the protected subpopulation (i.e., persons adopting disease-avoiding behaviors) are taken to be less likely to acquire or transmit disease by a factor $m_b$. In the model, the incubation period is divided into 5 stages. Movement from one stage to the next occurs with rate constant $k_L$ [20]. The fraction of exposed persons who never become symptomatic is represented by $f_A$. The fraction of symptomatic persons who progress to severe disease (and hospitalization or isolation at home) is represented by $f_H$ [25]. The fraction of persons with severe disease who recover is represented by $f_R$. Persons with asymptomatic disease leave the immune clearance stage of infection and recover with rate constant $c_A$ [26]. Persons with mild symptomatic disease recover with rate constant $c_I$ [27]. Persons with severe disease recover with rate constant $c_H$

[28]. The five adjustable parameters are $t_0$, $p_0$, $\lambda_0$, $\beta$, and $f_D$. The parameter $t_0$ refers to the start time of local sustained COVID-19 transmission; $p_0$ is the initial non-zero value of $P_\tau(t)$, the stationary fraction of the local population that is practicing disease-avoiding behaviors; $\lambda_0$ is the initial non-zero value of $\Lambda_\tau(t)$, a sum of rate constants that establishes a time scale for the establishment of the quasi-stationary state of NPIs; $\beta$ is the disease transmission rate constant (or contact rate parameter) in the absence of NPIs; and $f_D$ is the fraction of new infections detected in surveillance. The parameter $f_D$ characterizes the effectiveness of surveillance and relates new cases to new infections. In the model, $P_\tau(t)$ and $\Lambda_\tau(t)$ are taken to be step functions. Each of these functions has a value of 0 until $t = \sigma$ and thereafter changes value at a set of $n$ times (if $n > 0$), denoted $\tau = \{\tau_1 > \sigma, \ldots, \tau_n > \tau_{n-1}\}$. The value of $n$ starts at 0 and is incremented through model selection as described above. There is one additional adjustable parameter, $r$, the dispersion parameter of a negative binomial distribution $NB(p, r)$ used to characterize noise in case detection [7]. The value of $r$ is inferred jointly with the five adjustable model parameters.

In the case of one additional NPI period beyond the initial period ($n = 1$), three more adjustable parameters are used, which are denoted $\tau_1$, $p_1$, and $\lambda_1$. The latter two parameters determine the new values of $P_\tau(t)$, and $\Lambda_\tau(t)$ at time $t = \tau_1$, the start time of the second phase of NPIs. In general, three more adjustable parameters are added to the model each time $n$ is incremented. The equations of the compartmental model and of the auxiliary model can be found in Appendix 1 of Lin et al. [7].

Simulations were performed using ODE solvers available within the SUNDIALS software package [29].

## Data analysis

Bayesian inference was enabled by Markov chain Monte Carlo (MCMC) sampling, which yielded posterior samples for NPI parameters of region-specific compartmental mathematical models. Inferences were based on COVID-19 daily confirmed case count data available for the period starting on 21-January-2020 and ending on 14-September-2020. Inferences were conditioned on one to three periods of NPIs, uniform proper priors, a negative binomial model for surveillance noise, and estimates of selected parameters taken to have the same values across all regions of interest [7]. Following Lin et al. [7], we used model selection to determine the most parsimonious number of NPI periods. A model structure and parameterization were thus found for each region of interest. Each parameterization allows the model with selected structure to explain the corresponding regional epidemic curve.

We quantified uncertainty in daily case reports through resampling of the parameter posterior to generate a posterior predictive distribution for daily number of new cases detected [7].

We used an adaptive MCMC sampling algorithm described earlier [30] and implemented in the PyBioNetFit software package [31]. PyBioNetFit job setup files for the inferences performed in this study, including data files, are available online (https://github.com/lanl/PyBNF/tree/master/examples/Miller2022NavajoNation).

## Ethical considerations

This research is not considered human subjects research because we played no role in the collection of the data, the data does not include any identifying information, and the data were obtained from public sources.

## Results

The model we used to analyze data from the NN and surrounding states is illustrated in Fig 1. The model accounts for movement of persons between different states of protection against

SARS-CoV-2 infection because of disease-avoiding behaviors. In the model, persons are allowed to be in three states of protection: a state in which an uninfected person is protected imperfectly against infection because of disease-avoiding behaviors, a state in which an uninfected person is more exposed to infection because they do not take any special precautions to avoid infection, and a state in which an infected person is quarantined or in self-isolation. In the model, an initial NPI period ($n = 0$) begins as soon as the number of cumulative cases reaches or exceeds 200. A new NPI period is introduced through the model-selection procedure described in Methods. When a new NPI period is introduced, $n$ is incremented and NPI parameters change.

Figs 2 and 3 show 95% credible intervals of posterior predictive distributions for daily case detection for the NN and the four surrounding states. Posterior predictive distributions were found by drawing from parameter posterior samples generated through MCMC sampling, thereby propagating parametric uncertainty into prediction uncertainty. In the posterior predictive distributions, NN only has one surge in disease incidence whereas the surrounding states each have at least two surges.

Figs 2 and 3 show curves for the daily number of new symptomatic infections (vs. cases) based on maximum a posteriori (MAP) estimates for parameters (which are equivalent to maximum likelihood estimates because of the use of uniform proper priors). In our

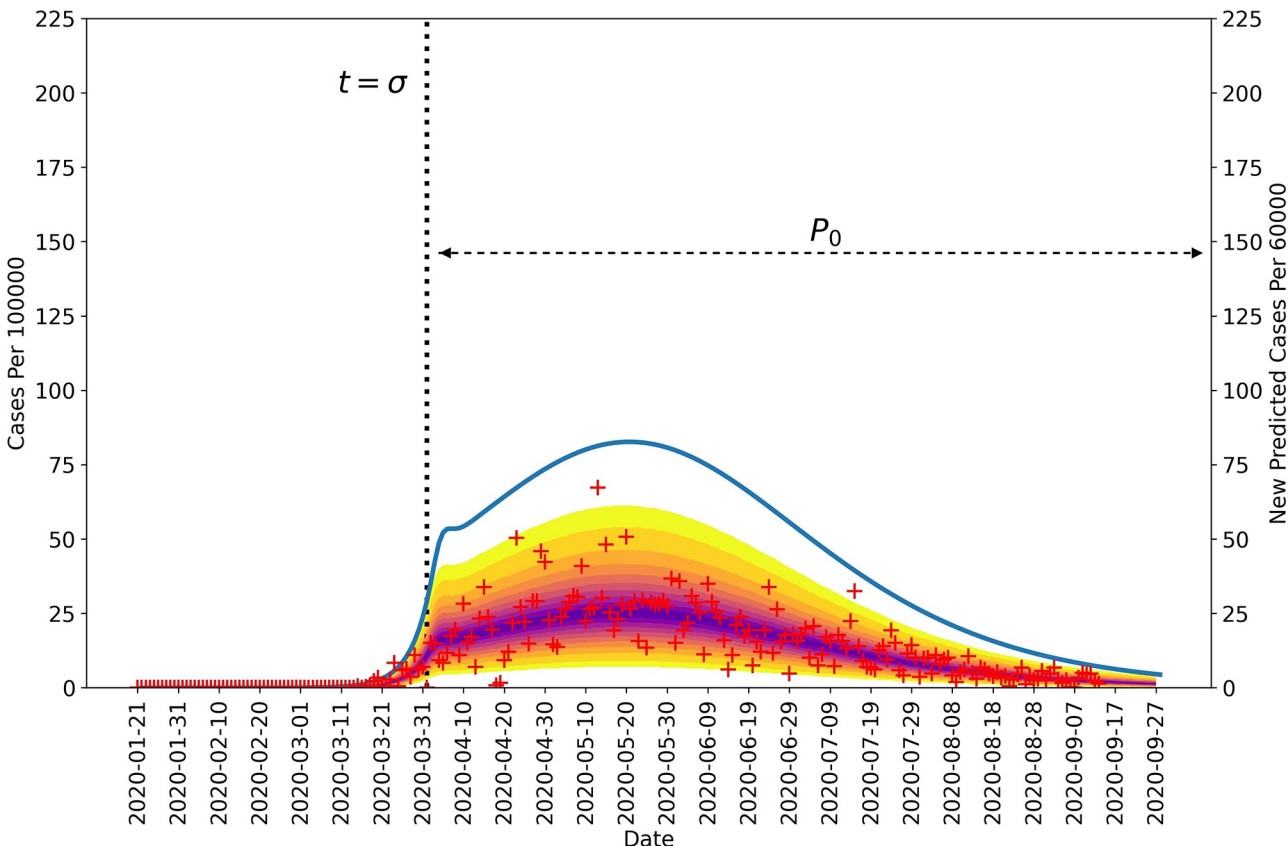

**Fig 2. Posterior predictive distribution for new cases detected in the Navajo Nation between 21-January-2020 and 14-September-2020.** The daily number of new COVID-19 cases detected in the Navajo Nation are indicated by red markers. The median percentiles of posterior samples are shown in purple. The blue curve indicates daily number of new infections and is based on MAP estimates for model parameters. The vertical black dotted line represents the time at which NPIs began in the Navajo Nation. The horizontal black dotted line indicates the duration of the initial NPI phase. It should be noted that the left and right vertical scales are different.

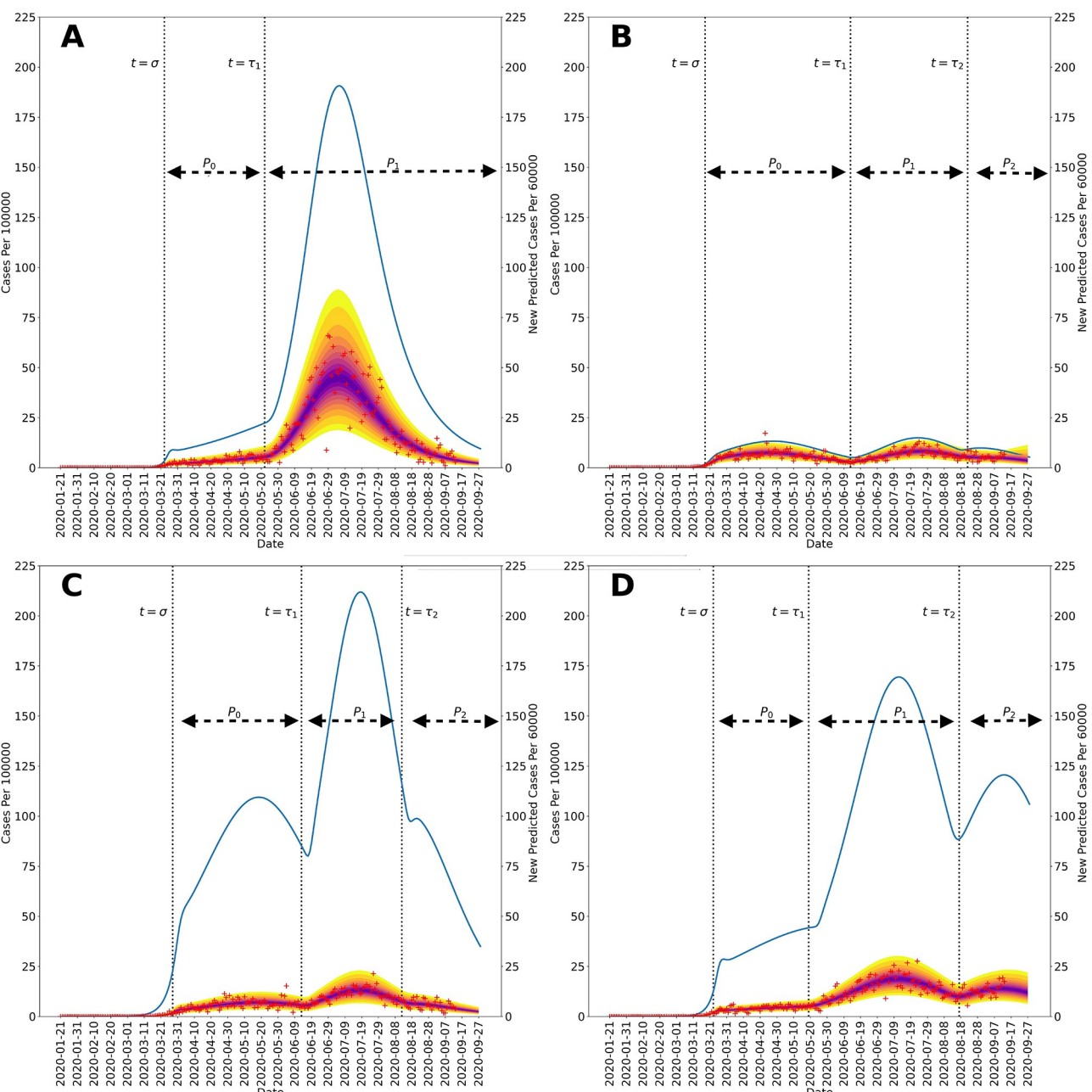

**Fig 3. Posterior predictive distributions for new cases in the four US states surrounding the Navajo Nation between 21-January-2020, and 14-September-2020: (A) Arizona, (B) Colorado, (C) New Mexico, and (D) Utah.** Recorded region-specific daily new cases of COVID-19 are indicated by red markers in each panel. The median parameter posterior estimates are shown in purple. The yellow bands delimited the 2.5 and 97.5 percentiles; the entire shaded region indicates the 95% credible interval. In each panel, the blue curve indicates daily number of new infections and is based on MAP estimates for region-specific model parameters. The start times of NPI phases are indicated by vertical dotted lines. The initial NPI phase begins when $t = \sigma$, the second NPI phase begins when $t = \tau_1$, and the third NPI phase begins when $t = \tau_2$. The horizontal black dotted lines indicate durations of NPI phases. It should be noted that the size of the first surge in Arizona, occurring in March and April 2020, is dwarfed by the size of the second surge. It should be noted that the left and right vertical scales of each panel are different.

calculations, the number of detected cases over a 1-d period is taken to be a fraction $f_D$ of the number of new symptomatic infections generated during that same period. The value of $f_D$ is region-specific. The MAP estimate for $f_D$ is 0.2 for the Navajo Nation, 0.15 for Arizona, 0.35 for Colorado, 0.04 for New Mexico, and 0.07 for Utah.

Figs 2 and 3 indicate when distinct NPI periods were determined to have begun. Table 1 summarizes results of the model-selection procedure used to decide between 1 or 2 or more NPI phases for each region of interest. The Navajo Nation was the only region of interest to have $\Delta AICc$ and $\Delta BIC$ values indicating only one NPI phase.

Fig 4 shows the marginal posteriors of the setpoint parameters $\{p_0, \ldots, p_n\}$ for each region, which were generated by MCMC sampling. Recall that each of these parameters determines the quasi-stationary population fraction adopting disease-avoiding behaviors and that there is a distinct setpoint for each distinct NPI phase (e.g., $p_0$, $p_1$, and $p_2$ for a region with three distinct NPI phases). For the Navajo Nation, we inferred only a single NPI phase over the period of interest. This phase is characterized by a NN-specific value for the setpoint parameter $p_0$. The marginal posterior for $p_0$ for the NN is shown in Fig 4A. In surrounding states, we inferred changes in adherence to disease-avoiding behaviors, i.e., different setpoints over time. The marginal posteriors for the state-specific setpoint parameters are shown in Fig 4B–4E.

Comparison of the marginal posteriors for different NPI phases within a given state reveals a relaxation in disease-avoiding behaviors in each state. Fig 5 shows MAP estimates for NPI setpoint parameters (e.g., $p_0$) for each region of interest over time. A higher setpoint indicates a higher prevalence of disease-avoiding behaviors. For the period of interest, we found that all regions experienced a decrease in their setpoint parameter values after an initial NPI phase except the Navajo Nation. Although Arizona, Colorado, and Utah initially had a higher setpoint than the Navajo Nation, the Navajo Nation maintained the initial setpoint for a longer period in comparison to the surrounding states.

Fig 6 shows Navajo Nation COVID-19 case data from 21-January-2020 to 5-February-2021 and projections of daily case counts for selected NPI scenarios after 14-September-2020. Between 21-January-2020 and 14-September-2020, the Navajo Nation maintained disease-avoiding behaviors (as characterized by the setpoint parameter $p_0$) and experienced only one surge in COVID-19 cases. However, after this period, the Navajo Nation experienced an additional surge in COVID-19 cases. The solid red curve in Fig 6 is the trajectory corresponding to the MAP estimate of $p_1$, obtained using data collected after 14-September-2020, and the dotted curves are different hypothetical trajectories based on lower and higher values for the NPI parameter $p_1$. We found that the Navajo Nation would have needed to maintain a value for $p_1$ greater than 0.27 after 14-September-2020 to avoid a surge in disease transmission.

Fig 7 presents a timeline of governmental mandates between 21-January-2020 and 14-September-2020 in the Navajo Nation and the four surrounding states. Four mandates are

**Table 1. Results from our model-selection procedure used to select the number of NPI periods in each region.**

| Region | $\Delta AICc$ | $\Delta BIC$ |
|---|---|---|
| Navajo Nation | -6.4 | -15.7 |
| Arizona | 76.2 | 66.7 |
| Colorado | 108.3 | 99.1 |
| New Mexico | 56.6 | 47.4 |
| Utah | 98.1 | 89.0 |

We calculated the value of the Akaike information criterion corrected for small sample size ($AICc$) for $n = 0$ and $n = 1$ versions of the model, as well as and value of the Bayesian information criterion ($BIC$) for the same two versions of each model. We defined $\Delta AICc = AICc^{n = 0} - AICc^{n = 1}$ and $BIC = BIC^{n = 0} - BIC^{n = 1}$. We adopted $n = 1$ over $n = 0$ when $\Delta AICc > 10$ and $\Delta BIC > 10$ (i.e., we reject the hypothesis that $n = 0$ when both $\Delta AICc$ and $\Delta BIC$ are greater than 10). Accordingly, $n = 0$ is indicated only for the Navajo Nation and $n > 0$ is indicated for all four surrounding states.

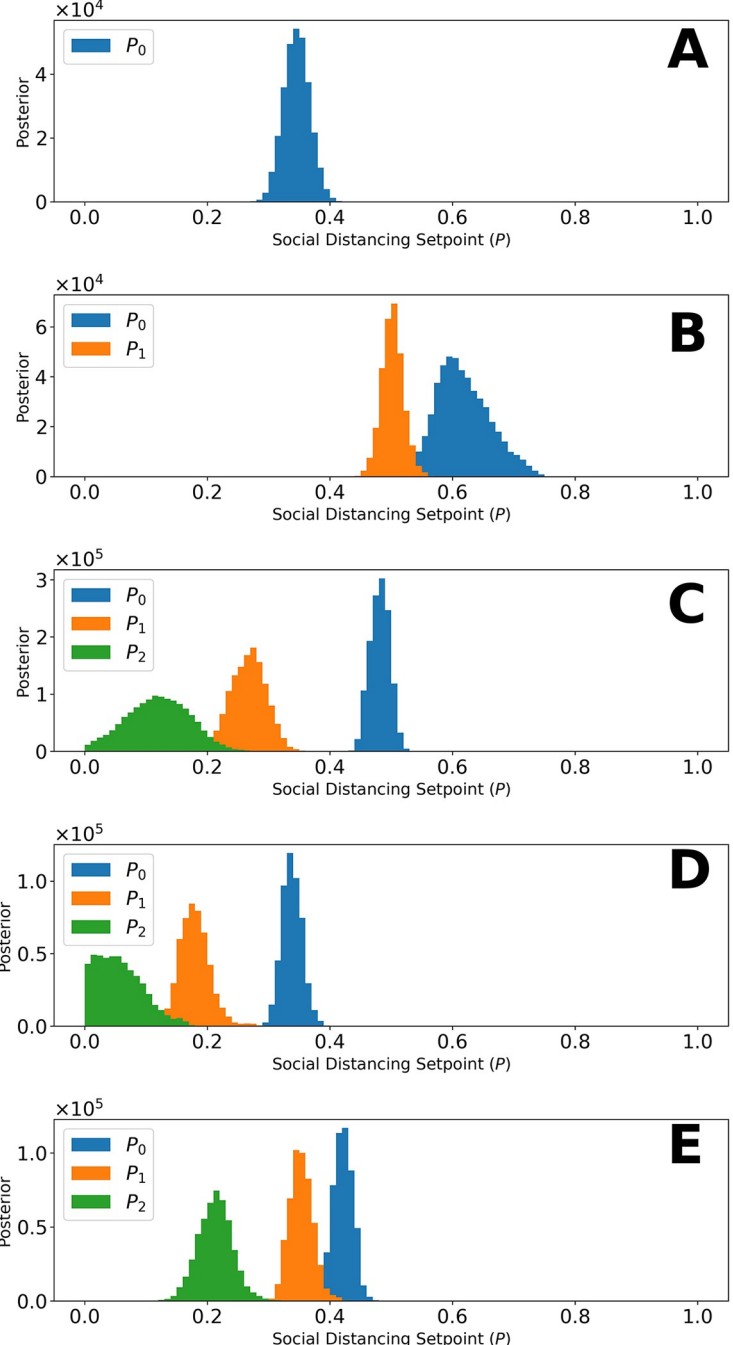

**Fig 4. Marginal posteriors for parameters of the setpoint function $P_\tau(t)$ (e.g., $p_0$) for (A) Navajo Nation, (B) Arizona, (C) Colorado, (D) New Mexico, and (E) Utah for the time period January 21, 2020, to September 14, 2020.** Recall that $P_\tau(t)$ denotes the fraction of the population practicing disease-avoiding behaviors at time $t$. The value of $P_\tau(t)$, a step function, is determined by one or more setpoint parameters, denoted $p_0$, $p_1$, etc. The Navajo Nation setpoint function parameter has the following maximum a posteriori (MAP) value: $p_0 = 0.35$. The Arizona setpoint function parameters have the following MAP values: $p_0 = 0.60$ and $p_1 = 0.5$. The Colorado setpoint function parameters have the following MAP values: $p_0 = 0.47$, $p_1 = 0.27$, and $p_2 = 0.11$. The New Mexico setpoint function parameters have the following MAP values: $p_0 = 0.34$, $p_1 = 0.19$, and $p_2 = 0.05$. The Utah setpoint function parameters have the following MAP values: $p_0 = 0.43$, $p_1 = 0.35$, and $p_2 = 0.21$ For each region of interest, the NPI switch times, $\tau = \{\tau_1, \ldots, \tau_n\}$, are indicated in Fig 3.

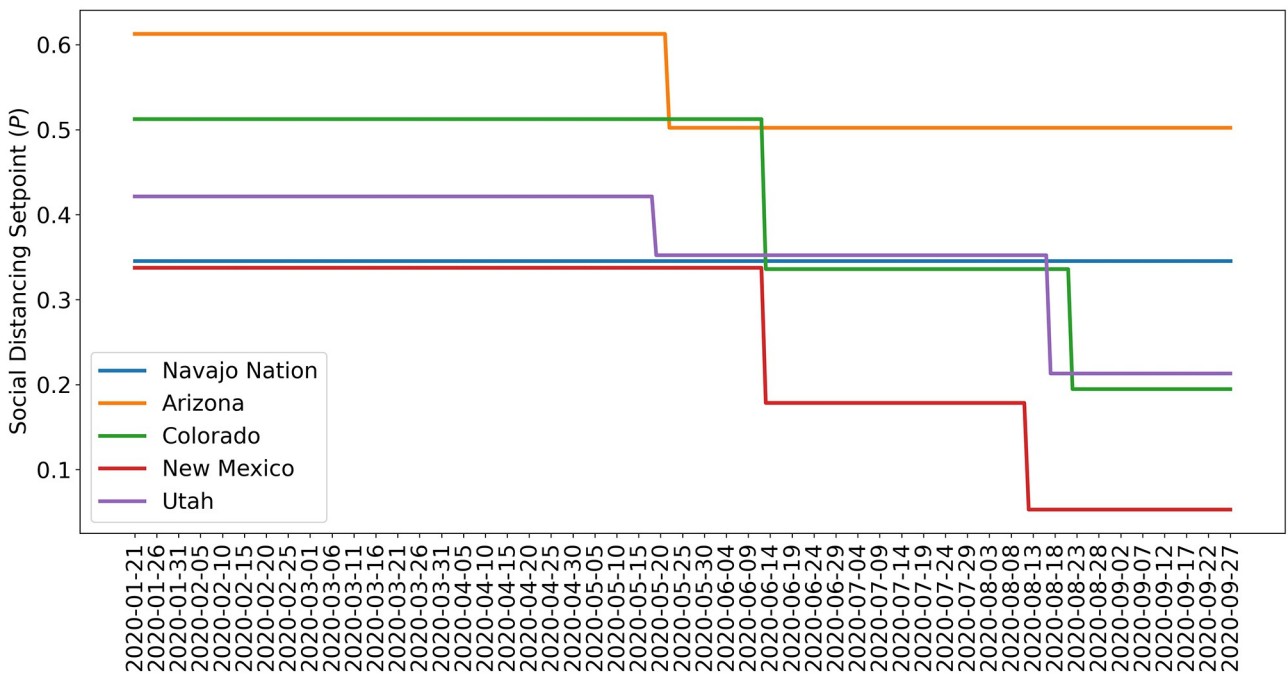

**Fig 5. Value of the NPI setpoint function $P_\tau(t)$ over time based on MAP estimates for NPI setpoint parameters $\{p_0, \ldots, p_n\}$ for (A) Navajo Nation (B) Arizona, (C) Colorado, (D) New Mexico, and (E) Utah.** The period considered is 21-January-2020 to 14-September-2020.

considered, which are related to face mask wearing, mass gatherings, non-essential business closures, and weekend lockdowns. As can be seen, these mandates were in effect for the longest duration in the Navajo Nation. Mandates in surrounding states were in effect for shorter durations and were imposed less consistently.

## Discussion

In this study, we used a compartmental model to quantify the overall effect of non-pharmaceutical interventions (NPIs) on COVID-19 transmission in specific regions, namely the Navajo Nation and the four surrounding states. In our model, we do not consider multiple strains of SARS-CoV-2 because during the time period of interest (21-January-2020 to 14-September-2020) variants of concern had yet to emerge. The first variant of concern (Alpha, B.1.1.7) appeared in the United Kingdom (UK) in late September of 2020 [32] and was detected in the US in November 2020, which is beyond the time window of our study [33].

The model for a given region includes a set of NPI setpoint parameters, each of which represents the quasi-stationary fraction of the regional population that is practicing disease-avoiding behaviors for a given period. By using surveillance data (daily case counts) to infer the region-specific values of the NPI setpoint parameters, we quantified the relative overall effectiveness of NPIs across the regions of interest.

From 21-January-2020 to 14-September-2020, we found that the Navajo Nation maintained the initial NPI setpoint throughout this period (Figs 4 and 5), consistent with a single surge in COVID-19 incidence (Fig 2). In contrast, we found that the surrounding states of Arizona, Colorado, New Mexico, and Utah did not. That is, each surrounding state had two or more NPI phases, marked by different NPI setpoints and multiple surges in COVID-19 incidence. These findings are consistent with a comparison of governmental mandates across the five

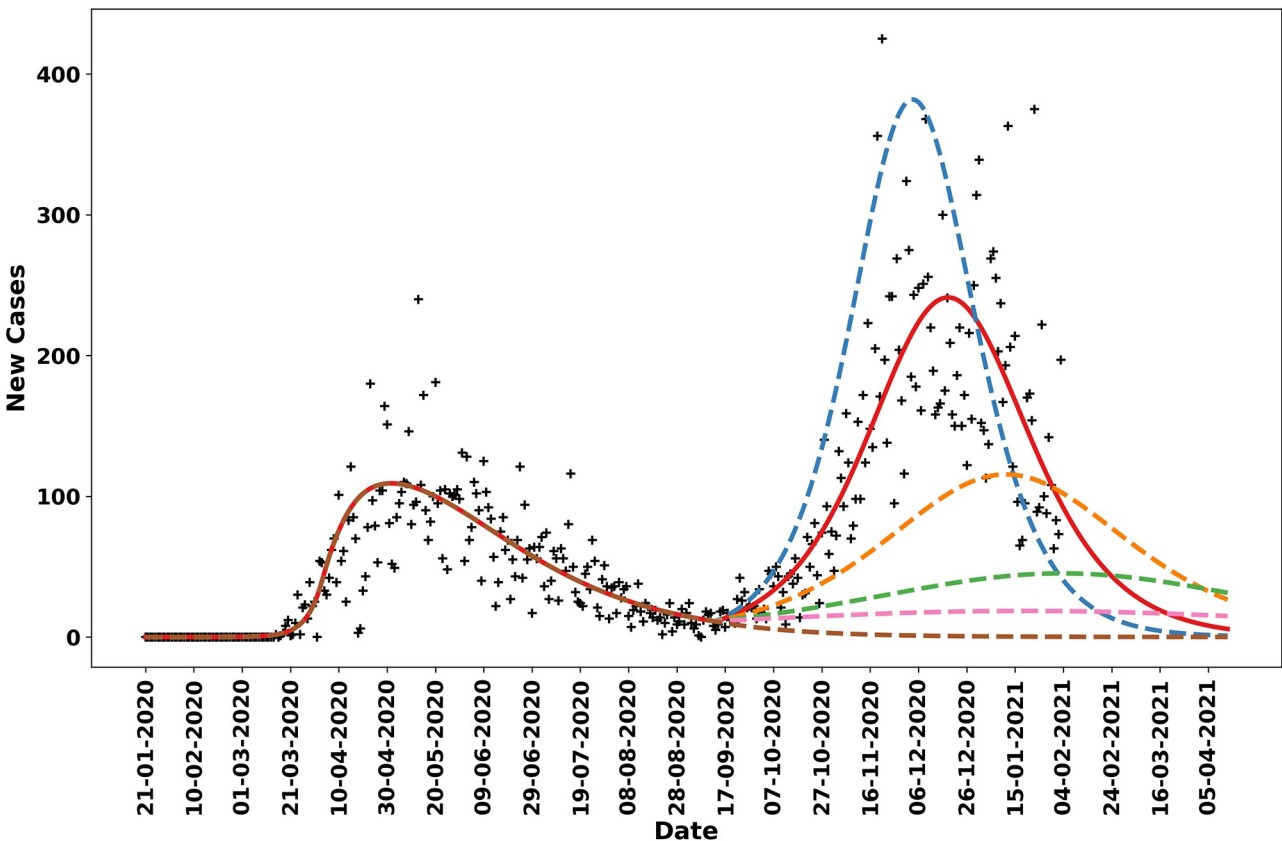

**Fig 6. Model-derived projections for various scenarios in which the NPI parameter $p_1$, which indicates the fraction of the population practicing disease-avoiding behaviors in a second NPI phase in the Navajo Nation starting 14-September-2020, was adjusted to identify the threshold required to prevent a second surge in cases.** The red solid line corresponds to the MAP estimate for $p_1$, which is approximately 0.19. The blue broken line indicates the predicted trajectory for daily cases when $p_1$ is fixed at 0.15. The orange broken line corresponds to a scenario wherein $p_1$ is fixed at 0.22, the green broken line corresponds to a scenario wherein $p_1$ is fixed at 0.25, the pink broken line corresponds to a scenario wherein $p_1$ is fixed at 0.27, and the brown broken line corresponds to a scenario wherein $p_1$ is fixed at 0.35.

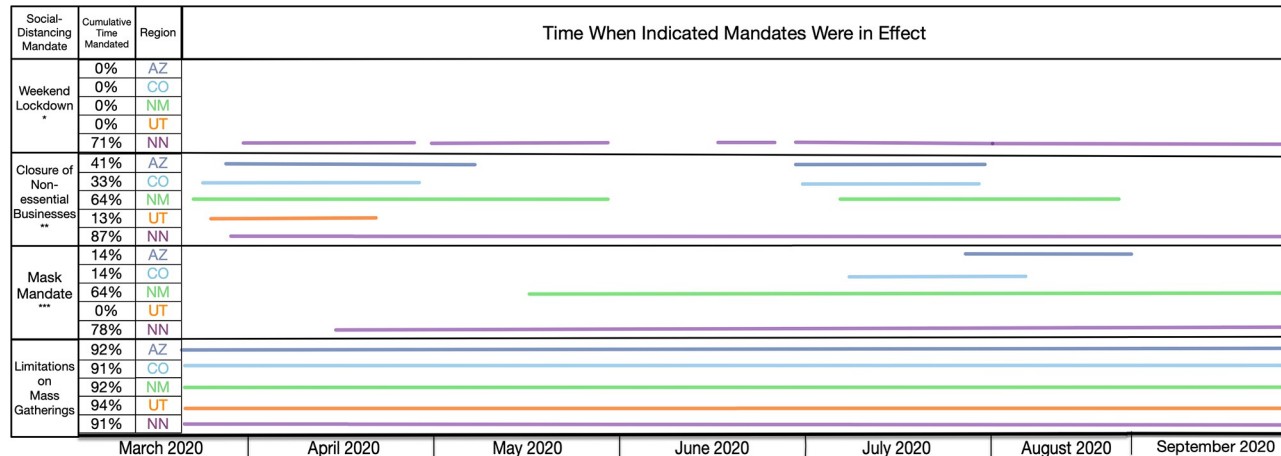

**Fig 7. Timeline for mandated NPIs in the Navajo Nation and surrounding states between 01-March-2020 and 14-September-2020.** Each region is represented by a different color, as indicated. Only mandates issued by state governors and the president of the Navajo Nation are considered.

regions of interest (Fig 7). Sustained NPIs is unique to the Navajo Nation and suggests an explanation for why this region experienced only one surge in COVID-19 cases while other regions experienced multiple surges. A generalizable result that we obtained is that sustained adherence to NPIs can prevent surges in disease incidence whereas relaxation of NPIs can lead to surges in disease. Our results add evidence in support of the idea that the duration of NPI adherence is an important factor influencing their effectiveness [34,35].

Interestingly, we inferred that the fraction of the NN population adopting disease-avoiding behaviors upon initial implementation of NPIs was lower than that in Arizona, Colorado, and Utah and similar to that in New Mexico. These results implicate sustained NPIs (rather than effectiveness of NPIs) as the reason for the different disease transmission dynamics between the Navajo Nation (only a single surge in disease incidence) and the surrounding states (multiple surges).

Consistent with other studies reviewed by Lison et al. [10], our results suggest that NPIs, even if only partially adopted, can slow and control disease transmission if mandates are consistent and are not relaxed prematurely or to too great an extent. We determined that the Navajo Nation's NPI setpoint parameter value between 21-January-2020 and 14-September-2020 was 0.35 but the value changed to 0.19 after 14-September-2020, preceding a second surge in disease incidence. We determined the NN minimum NPI setpoint parameter value needed to maintain control of disease transmission (i.e., to avoid a surge in disease incidence) to be 0.27 (Fig 6). In other words, the second surge in the NN could have been prevented if 27% of the population had maintained disease-avoiding behaviors after 14-September-2020.

We inferred two other notable differences between the regions of interest beyond differences in adoption of effective NPIs. First, surveillance efforts may have had different levels of effectiveness. Our MAP estimates for $f_D$, the fraction of new infections detected, ranged from a low of 0.04 for New Mexico to a high of 0.35 for Colorado. Colorado, New Mexico, and Utah had similar numbers of cases per 100,000 residents but the inferred differences in surveillance effectiveness suggest that COVID-19 impacts were significantly greater in New Mexico and Utah than in Colorado. Second, there were differences in contagiousness across the regions of interest. Our MAP estimates for $\beta$, the contact rate parameter, ranged from just over 0.3 per day for the Navajo Nation and New Mexico to just over 0.5 per day for Arizona. Using the formula for the basic reproduction number $R_0$ given by Mallela et al. [9], these differences in $\beta$ estimates translate into the following estimated $R_0$ values for the five regions: 3.6 for New Mexico, 3.7 for the Navajo Nation, 4.4 for Utah, 4.6 for Colorado, and 5.9 for Arizona. Thus, the relatively high adoption of effective NPIs in Arizona was offset by relatively high transmission of COVID-19. Our analysis does not provide insight into why contagiousness varied across the regions of interest.

We detected changes in disease-avoiding behaviors over time using a model selection procedure, which indicates when an NPI setpoint needs to change value for consistency with surveillance data. Our approach for detecting changes in disease-avoiding behaviors has at least two limitations. First, we cannot ascertain the relative effectiveness of individual NPIs. The reason is that our model only accounts for the overall effect of all NPIs. In the model, persons are either protected by NPIs or not. Second, our model can only explain surges in disease incidence caused by relaxation of NPIs. The model does not account for other factors that could cause surges, such as an increased disease transmissibility associated with emergence of a viral variant of concern or loss of immunity. Variants of concern did not emerge during the period of interest [32,33]. Sterilizing immunity against SARS-CoV-2 infections wanes but reinfection was unlikely to contribute to disease incidence during the period of interest [36].

In summary, our analysis suggests that once NPIs have brought an outbreak under control, relaxation of the NPIs can be implemented but relaxation should be measured to avoid a new

surge in disease incidence. A relatively low level of disease incidence is not an indicator that NPIs can be safely relaxed. In future pandemics a model accounting for NPIs could perhaps be used to guide relaxation of NPIs [37].

## Author Contributions

**Conceptualization:** Ely F. Miller, William S. Hlavacek, Richard G. Posner.

**Data curation:** Ely F. Miller.

**Funding acquisition:** Ye Chen, Abhishek Mallela, Yen Ting Lin, William S. Hlavacek, Richard G. Posner.

**Investigation:** Ely F. Miller.

**Methodology:** Ely F. Miller, Jacob Neumann, Abhishek Mallela, Yen Ting Lin, William S. Hlavacek.

**Project administration:** William S. Hlavacek, Richard G. Posner.

**Resources:** Richard G. Posner.

**Software:** Ely F. Miller, Jacob Neumann, Yen Ting Lin.

**Supervision:** Ye Chen, William S. Hlavacek, Richard G. Posner.

**Validation:** Ely F. Miller, Abhishek Mallela.

**Visualization:** Ely F. Miller.

**Writing – original draft:** Ely F. Miller, William S. Hlavacek, Richard G. Posner.

**Writing – review & editing:** Ely F. Miller, Jacob Neumann, Ye Chen, Abhishek Mallela, Yen Ting Lin, William S. Hlavacek, Richard G. Posner.

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
