## [Decision Letter · Decision Letter 0]

18 Apr 2023

PGPH-D-22-02023

Quantification of early nonpharmaceutical interventions aimed at slowing transmission of Coronavirus Disease 2019 in the Navajo Nation and surrounding states (Arizona, Colorado, New Mexico, and Utah)

Dear Dr. Posner,

Thank you for submitting your manuscript to PLOS Global Public Health. After careful consideration, we feel that it has merit but does not fully meet PLOS Global Public Health’s publication criteria as it currently stands. Therefore, we invite you to submit a revised version of the manuscript that addresses the points raised during the review process.

We look forward to receiving your revised manuscript.

Kind regards,

Miguel Angel Garcia-Bereguiain, PhD

Academic Editor

Journal Requirements:

1. Please amend your detailed online Financial Disclosure statement. This is published with the article. It must therefore be completed in full sentences and contain the exact wording you wish to be published.

a) State the initials, alongside each funding source, of each author to receive each grant. For example: "This work was supported by the National Institutes of Health (####### to AM; ###### to CJ) and the National Science Foundation (###### to AM)."

2. Please update your online Competing Interests statement. If you have no competing interests to declare, please state: “The authors have declared that no competing interests exist.”

3. We ask that a manuscript source file is provided at Revision. Please upload your manuscript file as a .doc, .docx, .rtf or .tex.

4. Please remove any figures embedded in your manuscript file, leaving only the individual TIFF/EPS image files. You may leave the figure captions or legends in the manuscript.

Additional Editor Comments (if provided):

Reviewers' comments:

Reviewer's Responses to Questions

**Comments to the Author**

1. Does this manuscript meet PLOS Global Public Health’s publication criteria? Is the manuscript technically sound, and do the data support the conclusions? The manuscript must describe methodologically and ethically rigorous research with conclusions that are appropriately drawn based on the data presented.

Reviewer #1: Partly

Reviewer #2: Partly

2. Has the statistical analysis been performed appropriately and rigorously?

Reviewer #1: Yes

Reviewer #2: Yes

3. Have the authors made all data underlying the findings in their manuscript fully available (please refer to the Data Availability Statement at the start of the manuscript PDF file)?

Reviewer #1: Yes

Reviewer #2: Yes

4. Is the manuscript presented in an intelligible fashion and written in standard English?

Reviewer #1: Yes

Reviewer #2: Yes

5. Review Comments to the Author

Reviewer #1: In the summary part, the IMRaD structure must be respected. Break down each part. The main objective of the study should be clearly stated. If possible, the main results in terms of quantification of the measures, which really show the measured impact of the interventions, should also be included.

I would therefore like to see the whole summary taken up again, taking into account the comments.

The introduction failed to address the general aspect of the subject. It would therefore be better to shed light on the situation of approaches that have approached it in the same way in other countries outside your study framework. As a result, your discussion has stopped at mere comments. There is really no discussion under the subheading discussion of results with those of other authors discussion.

It would be really interesting to compare your work with what other authors have done elsewhere. If not present, mention it too.

Do not mix the methods and introduction sections

Why, is it directly in the results section that the objective of the study was first stated? Is it a result? I don't think so. I would like to see it brought back to the introduction of the abstract part and the end of the introduction part of the body of the article.

Reviewer #2: Comments

Dear authors have submitted a manuscript entitled "Quantification of early nonpharmaceutical interventions aimed at slowing transmission of Coronavirus Disease 2019 in the Navajo Nation and surrounding states (Arizona, Colorado, New Mexico, and Utah)". This manuscript is the product of Bayesian inference-based modeling and reports a very interesting finding that the effectiveness of NPIs lies in the fact that they are sustained measures rather than measures per se. This is very interesting research, however, there are limitations, especially methodological ones, which should be corrected before the manuscript is accepted.

Introduction

In general, the introductory section is well written and groups the most important aspects to provide context to the research.

However, in the last paragraph, between lines 93 and 103, the objective of the research is stated, and a brief "summary" of the methodology used is added. In the context of the form of the manuscript this is not correct. Please replace the text from line 93 to 103 with the proposed overall research objective of the manuscript.

Methodology

Dear authors, the materials and methods section, while describing useful and important information for the manuscript, has important deficiencies that must be addressed before the manuscript can be accepted for publication.

Although this is a modeling-based study, it is recommended that the section be restructured according to STROBE guidelines as soon as the methodology allows, including the sections on:

Study design

Setting

Exposure

Outcome

Data sources

Model calculation and modeling process

For example, the text between line 105 and 125 should be added to the "Data sources" section.

The description of the modeling performed is quite complete and I consider that the analysis methods are correct, however, the description as with the rest of the text in the methodology section should be structured in the corresponding section "Model calculation and modeling process". Please, in cases where equations are used make sure that these are described after an end point using the corresponding function to provide greater understanding to the reader.

As mentioned above, please make sure to add or restructure the necessary information to cover the mentioned sections.

Results

Please move the research objective to the end of the Introduction section.

Likewise, move the text from lines 208 to 213 to the corresponding section in the Methods section.

For the rest, the findings proposed in the results section should be objective to avoid confusion among readers, the figures are adequate and of quality I consider that they do not require changes.

Discussion

Line 282 to 289: Initially the limitations of the study should be mentioned in the final paragraph of the discussion section. On the other hand, they claim the limitation of not being able to determine the relative efficacy of NPIs equally, please add a logical argument for this limitation.

Also, while it is true that they mention that the model does not consider other variables that could make it more accurate, this is a correct assertion. However, regarding the possible association with the emergence of new SARS-CoV-2 variants, despite not being able to perform complex statistical analyses, you could conduct a descriptive analysis of variant circulation in the Navajo Nation and surrounding cities to weigh your findings against variants that were shown to have higher transmissibility rates than the native variant such as the Omicron variant. If you do not have access to this information, please add a detailed description of your limitation in the appropriate section.

6. PLOS authors have the option to publish the peer review history of their article (what does this mean?). If published, this will include your full peer review and any attached files.

**Do you want your identity to be public for this peer review?** For information about this choice, including consent withdrawal, please see our Privacy Policy.

Reviewer #1: **Yes: **Nicolas Hamondji AMEGAN

Reviewer #2: **Yes: **Juan S. Izquierdo-Condoy

---

## [Editor Report · Decision Letter 1]

30 May 2023

Quantification of early nonpharmaceutical interventions aimed at slowing transmission of Coronavirus Disease 2019 in the Navajo Nation and surrounding states (Arizona, Colorado, New Mexico, and Utah)

PGPH-D-22-02023R1

Dear Dr Posner,

We are pleased to inform you that your manuscript 'Quantification of early nonpharmaceutical interventions aimed at slowing transmission of Coronavirus Disease 2019 in the Navajo Nation and surrounding states (Arizona, Colorado, New Mexico, and Utah)' has been provisionally accepted for publication in PLOS Global Public Health.

Best regards,

Miguel Angel Garcia-Bereguiain, PhD

Academic Editor